# Testing the Resource Hypothesis of Species–Area Relationships: Extinction Cannot Work Alone

**DOI:** 10.3390/microorganisms10101993

**Published:** 2022-10-09

**Authors:** Wei Deng, Li-Lei Liu, Guo-Bin Yu, Na Li, Xiao-Yan Yang, Wen Xiao

**Affiliations:** 1Institute of Eastern-Himalaya Biodiversity Research, Dali University, Dali 671003, China; 2Collaborative Innovation Center for Biodiversity and Conservation in the Three Parallel Rivers Region of China, Dali 671003, China; 3The Provincial Innovation Team of Biodiversity Conservation and Utility of the Three Parallel Rivers Region, Dali University, Dali 671003, China; 4International Centre of Biodiversity and Primates Conservation, Dali 671003, China; 5Yunling Black-and-White Snub-Nosed Monkey Observation and Research Station of Yunnan Province, Dali 761003, China

**Keywords:** resources, extinction, dispersal, microbial diversity, species–area relationships

## Abstract

The mechanisms that underpin the species–area relationship (SAR) are crucial for both the development of biogeographic theory and the application of biodiversity conservation. Since its origin, the resource hypothesis, which proposes that rich resources in vast ecosystems will lower extinction rates and shape the SAR, has not been tested. The impossibility to quantify resources and extinction rates using plants and animals as research subjects, as well as the inability to rule out the influences of the area per se, habitat diversity, dispersal, and the historical background of biodiversity, make testing this hypothesis problematic. To address these challenges and test this hypothesis, two sets of microbial microcosm experimental systems with positive and negative correlated resources and volumes were created in this work. The results of 157 high-throughput sequencing monitoring sessions at 11 time points over 30 consecutive days showed that neither of the experimental groups with positive or negative correlations between total resources and microcosm volume had a significant SAR, and there were no negative correlations between extinction rates and resources. Therefore, in our microcosmic system, resources do not influence extinction rates or shape the SAR. Dispersal should be the principal mode of action if the resource theory is correct.

## 1. Introduction

The species–area relationship (SAR), which is the theoretical underpinning of island biogeography and an essential tool for biodiversity, as well as protected area designation, measures the positive link between species richness and habitat area [1,2,3,4,5]. The area per se hypothesis (large habitats have lower stochastic extinction rates) [3,6,7,8], the passive sampling hypothesis (large habitats gather more species from the species pool) [9], and the habitat diversity hypothesis (large habitats have more diverse habitats) [10] are the key ideas that describe the mechanics of SAR creation, in addition to resource hypothesis and edge effects, all of which are assumed to have a role in shaping the SAR [11].

According to the resource hypothesis of population ecology, population density rises as resource abundance rises, explaining the positive relationship between population density and habitat area [12,13,14]. According to Connor and McCoy (2013), an increase in resource abundance has the potential to shape the SAR by increasing population density of each species, and thus decreasing extinction rates [11]. To properly shape the SAR, this hypothesis needs one requirement and two sets of relationships. A greater habitat area with more resources is a requirement. In addition, there are two sets of relationships, including the resource–density relationship, in which more resources result in an increase in the population density of each species and the density–richness relationship, in which a rise in population density results in a decline in stochastic extinction rates, leading to the emergence of more species (Figure 1).

The resource–density relationship has recently been supported by more and more evidences, although the density–richness relationship is still less clear-cut. The resource hypothesis, therefore, has yet to be put to the test in terms of shaping SAR. The resource hypothesis confronts three key obstacles in its validity. First, it is difficult to distinguish the role of resources from the role of area per se. SAR is shaped by both resources and area via extinction [15,16], and an increase in natural habitat area typically entails a rise in resources. Second, the quantity of resources in vast habitats may entice more species to settle there, influencing SAR [17]. However, distinguishing the separate contributions of extinction and dispersal is challenging, since no natural community contains solely extinction or dispersal processes. Third, determining resource abundance, extinction, and dispersion rates is challenging, and previous research can only approximate them using models [18,19,20,21,22,23].

Based on this, experiments must be planned to overcome the three problems raised above in order to test the resource hypothesis for shaping SAR and must investigate the following: (1) whether habitats with more resources have a higher population density of each species, and (2) whether habitats with more resources have lower extinction rates that shape the SAR, while satisfying the premise that larger habitats have more resources. To address these challenges, two sets of microcosm experiments (consisted of sterile culture flasks, dry matter, and pao cai microbes) were established to test the resource hypothesis for shaping the SAR. The airtight, homogeneous, and consistent system of microbial origin ensures that influences such as dispersal, differences in biodiversity background [24], and habitat diversity are completely excluded in the microcosms. To rule out the effect of area per se, we conducted two groups of experiments in which microorganisms were continuously monitored using high-throughput sequencing techniques, with dry matter mass increasing with volume in the P group (positive group) and decreasing with volume in the N group (negative group).

There are four possible results from the experiments, which are as follows: (1) if both experimental groups have an SAR, the system’s SAR is completely determined by the area per se; (2) if neither experimental group has an SAR, extinction cannot independently shape the SAR, owing to resource disparities, and dispersion should be the more relevant process; (3) if group P has an SAR but group N does not, the SAR is only affected when the total resources is positively connected with area; (4) if group P does not have an SAR but group N does, neither resources nor area are the key variables that form SAR. Regardless of whether our results confirm the resource hypothesis, the outcomes of this work will offer fresh perspectives on SAR research thanks to the experimental approach used.

## 2. Materials and Methods

The experimental design and the process of establishing the microcosm system for this study are shown in Figure 2. When microcosms were created, culture flasks were filled with a certain amount of dry matter, sterile water was added, and the whole microcosm was sterilized. Following the addition of the microbial seed solution, each microcosm was cultured for 30 days at a constant temperature. Every three days throughout this time, a collection of samples was collected for high-throughput sequencing in order to collect data on microbial diversity (fungi and bacteria). Each sample’s pH, TN, TP, NO_3_/NO_2_, and PO_4_ contents were also determined.

### 2.1. Preparation of Microbial Seed Solution for Pao Cai

Pao cai (salted fermented vegetables) is a traditional fermented vegetable product that is high in microbial diversity and typically uses a variety of vegetables as raw materials. The vegetables are fermented in an anaerobic jar with a special pao cai brine.

We combined 3.5 kg white radish (*Raphanus sativus*), 3.5 kg cabbage (*Brassica oleracea*), 0.2 kg chilli (*Capsicum frutescens*), 0.1 kg ginger (*Zingiber officinale*), 0.1 kg Chinese-pepper (*Zanthoxylum bungeanum*), 0.25 kg rock sugar and 2 kg cold boiled water (containing 6% salt) and divided the solution into earthenware pao cai jars. After 7 days of natural fermentation at room temperature, the pao cai was filtered through sterile gauze to obtain 2000 mL of pao cai microbial seed solution, which was mixed well using a vortex shaker and set aside.

### 2.2. Preparation of Dry Matter

We washed and diced 50 kg of white radish, then dried it in a blast drying oven at 60 °C for 48 h.

### 2.3. Establishment of the Microcosmic System

The positive group was a microcosmic system in which the resource increased with volume. First, eleven 50 mL, 100 mL, 150 mL, 300 mL, 500 mL, 700 mL and 1000 mL culture flasks were taken and 0.5 g, 1.0 g, 1.5 g, 3.0 g, 5.0 g, 7.0 g and 10 g of dry matter were added in that order and distilled water was added to the flasks (leaving a volume greater than 1 mL for the addition of the microbial seed solution).

The negative group was a microcosmic system in which the resource decreased with volume. First, eleven 50 mL, 100 mL, 150 mL, 300 mL, 500 mL, 700 mL and 1000 mL culture flasks were taken and 10 g, 7.0 g, 0.5 g, 5.0 g, 3.0 g, 1.5 g and 1.0 g of dry matter were added in that order and distilled water was added to the flasks (leaving a volume greater than 1 mL for the addition of the microbial seed solution).

The above culture flasks were autoclaved at 121 °C for 30 min, cooled and 1 mL of pao cai microbial seed solution was added to each flask, replenished with sterile water until the flask was full and the flask was sealed without leaving any air inside. Each culture flask becomes a microcosm system. We placed each microcosm system in a constant temperature, light-proof incubator at 25 °C for 30 days.

### 2.4. Sample Collection and Determination of Environmental Factors

Prior to the formation of the microcosm, one sample of well-mixed pao cai soup was used as a reference. After the microcosm system was constructed, samples were collected on days 1, 3, 6, 9, 12, 15, 18, 21, 24, 27, and 30, with one sample from each volume of the positive and negative groups taken at each sampling. Over the course of 30 days, 155 samples were collected at 11 different time points. After thoroughly mixing the microcosm system, 50 mL of culture fluid was taken as the sample, centrifuged at 8000 rpm for 10 min, and stored at −80 °C for precipitation. A segmented flow analyzer was used to determine the concentrations of total nitrogen (TN-mg/L), total phosphorus (TP-mg/L), orthophosphate content (PO_4_-mg/L), and nitrate/nitrite content (NO_3_/NO_2_-mg/L) (Seal Analytical AA3, Berlin, Germany). Using a pH meter, the pH level of each sample was determined.

### 2.5. Microbial Analyses

First, 16S and ITS amplicon sequencing of the samples was carried out using second generation sequencing technology. The sequencing service were provided by Wekemo Tech Group Co., Ltd., Shenzhen, China.

Microbial DNA was extracted from the samples using the E.Z.N.A.^®^ Soil DNA Kit (Omega Bio-tek, Norcross, GA, USA), according to the manufacturer’s protocols. For bacteria, we targeted the V3-V4 regions of the 16S ribosomal RNA (rRNA) gene using the 338F (5′-ACTCCTACGGGAGGCAGCAG-3′) and 806R (5′-GGACTACHVGGGTWTCTAAT-3′) primer pairs [25]. For fungi, we targeted the ITS1-1F regions of the nuclear ribosomal internal transcribed spacer region (ITS rDNA) gene, using the ITS1-1F-F (5′-CTTGGTCATTTAGAGGAAGTAA-3′) and ITS-1F-R (5′-GCTGCGTTCTTCATCGATGC-3′) primer pairs [26]. PCRs were performed in triplicate in a 20 μL mixture that contained 4 μL of 5× FastPfu Buffer, 2 μL of 2.5 mM dNTPs, 0.8 μL of each primer (5 μM), 0.4 μL of FastPfu Polymerase and 10 ng of template DNA. The PCR program for the 16S rRNA gene was as follows: 3 min of denaturation at 95 °C, 27 cycles of 30 s at 95 °C, 30 s of annealing at 55 °C, 45 s of elongation at 72 °C, and a final extension at 72 °C for 10 min. For the ITS1-1F region, the PCR program was as follows: initial denaturation at 98 °C for 1 min, 30 cycles of denaturation at 98 °C for 10 s, primer annealing at 50 °C for 30 s, and extension at 72 °C for 30 s. A final extension step of 5 min at 72 °C was added to ensure complete amplification of the target region. The resulting PCR products were extracted from a 2% agarose gel and further purified using the AxyPrep DNA Gel Extraction Kit (Axygen Biosciences, Union City, CA, USA) and quantified using QuantiFluor™-ST (Promega, Madison, WI, USA), according to the manufacturer’s protocols.

Purified amplicons were pooled in equimolar amounts and paired-end sequenced (2 × 300) on an Illumina NovaSeq platform (Illumina, San Diego, CA, USA), according to standard protocols. The analysis was conducted by following the “Atacama soil microbiome tutorial” in Qiime2docs, along with customized program scripts (https://docs.qiime2.org/2019.1/, accessed on 5 March 2022). Briefly, FASTQ files that contained the raw data were imported into the QIIME format with the QIIME2 system, using the QIIME tools import program. Demultiplexed sequences from each sample were quality filtered and trimmed, de-noised, and merged, and then the chimeric sequences were identified and removed using the QIIME2 DADA2 plugin to obtain the feature table of amplicon sequence variants (ASVs) [27,28]. An ASV is any one of the inferred single DNA sequences recovered from a high-throughput analysis of marker genes. Because these analyses, also called “amplicon reads,” are created following the removal of erroneous sequences generated during PCR and sequencing, using ASVs makes it possible to distinguish sequence variation by a single nucleotide change. ASVs have been proposed as an alternative to operational taxonomic units (OTUs) for analyzing microbial communities. The QIIME2 feature-classifier plugin was then used to align ASV sequences to a pre-trained GREENGENES 13_8 99% database (trimmed to the V3V4 region bound by the 338F/806R primer pair, for bacteria) and UNITE database (for fungi) to generate the taxonomy table [29].

### 2.6. Data Analyses

To determine if the microcosm system was set up in line with the pre-defined objectives, microbial diversity profiles were initially analyzed. Alpha and beta diversity were determined for each sample using the microeco and vegan packages in R 4.20 [30,31]. To show the diversity and structure of microbial community, plot stacked histograms were used and we conducted NMDS analysis of the microbial community based on Bray–Curtis distances. R 4.20 was used to determine the extinction rate by comparing the number of species extinct in each system to the reference sample, which consisted of three samples taken on the day the microcosm was created. R 4.20 was used to depict the linearity of microbial species richness and extinction rates with time for the microcosm systems.

Second, the correlation between resource abundance and population densities was examined to investigate whether greater resources could shape higher population densities; the correlation between extinction rates and volume, resources, and environmental factors was examined to investigate whether greater resource abundance could lessen extinction rates; and finally, SAR curves were built to investigate whether the resource hypothesis could influence SAR. The population density of each species of microorganism in high-throughput sequencing data is equivalent to the relative abundance of each species. The Kernel density estimate graph of the resources and relative abundance of each taxon was created using the ggplot2 package of R. The complete disappearance of an ASV is recorded as extinction. A mental test was conducted and a correlation graph for environmental factors and fungal species richness, bacterial species richness, and total extinction rate was created using the ggcor and ggplot2 packages [32]. SAR curves were plotted and tested for significance using a linear model in log-log space, using culture bottle volumes instead of areas.

Finally, by utilizing microbial interaction network analysis, the influence of resources on microbial interactions in microcosmic systems was investigated. The mapping of microbial networks and the computation of network properties were carried out using the R package ggClusterNet [33].

## 3. Results

### 3.1. Overview of the Diversity and Extinction Rates of Microbes

This study yielded a total of 13,159,681 16S rRNA gene sequences, which were then quality filtered, trimmed, and de-noised to yield 9781,102 high quality sequences, totaling 12,587 bacterial ASVs. A total of 11,386,587 ITS gene sequences were obtained and after quality filtering, trimming, and de-noising the sequences, a total of 8,678,214 high quality sequences were obtained, yielding a total of 10,433 fungal ASVs. The ASVs monitored covered 3 domains, 61 phyla and 1211 genera of microbial taxa.

*Firmicutes*, *Proteobacteria*, *Bacteroidetes*, *Fusobacteria*, and *Actinobacteria* were the prominent bacterial phyla that co-occurred in the samples, while *Ascomycota*, *Basidiomycota*, *Mortierellomycota*, and *Rozellomycota* were the dominant fungal phyla that co-occurred in the samples (Figure 3). The NMDS analysis revealed that, while the communities in the two treatment groups separated, the overlap was greater (Figure 4, stress = 0.1641); the differences within the negative group were larger. Samples from the negative group contributed significantly to the differences between the two treatment groups.

In both the overall positive and negative groups, species richness in the microcosm system decreased considerably as the microcosm establishment time grew, and extinction rates increased drastically as the microcosm establishment time increased (Figure 5).

### 3.2. Resources Cannot Shape Microbial SAR

Mantel test analysis showed that resources, volume and all five environmental factors had no significant effect on extinction rates (Figure 6). TN and TP had a highly significant effect on bacterial species richness and pH had a highly significant effect on fungal species richness. NO_3_/NO_2_ and PO_4_ had a significant effect on bacterial species richness (Figure 6).

In both treatment groups, microbial taxa with high, medium and low relative abundance occurred with high frequency in the less resourceful microcosms (Figure 7).

Within 30 days of the microcosm system, no significant SAR was observed in any of the treatment groups when the dry matter mass was comparable to the microcosm volume (Figure 8). The statistical findings of each SAR curve are shown in Table 1.

### 3.3. Resources Do Not Alter the Structure and Complexity of Microbial Interactions Networks in the Microcosm

Both treatment groups shared a similar network structure for microbial interactions, with the dark red module serving as the primary one across all of them. The remaining modules are less connected, showing that the microbial communities are divided into several ecological niches, with the microbes in those ecological niches creating more autonomous network modules (Figure 9). The complexity of the microbial contacts network is reflected in the number of edges in the network. However, neither the volume nor the resources for the positive nor negative groups showed a link with network complexity (Figure 10).

## 4. Discussion

The microcosmic system was extremely diverse, with a total of 23,020 microbial ASVs found throughout the experimental system, and the taxa monitored were greater than in prior pao cai microbiological investigations [34,35,36]. At both the phylum and genus levels, the microorganisms we caught were consistent with the dominating taxa observed in earlier pao cai investigations [37,38,39]. As a result, our work gives a more thorough picture of pao cai microbial diversity than earlier studies, and our results are unaffected by sampling effects, due to the smaller experimental setup and high-resolution ASV. Simultaneously, the extinction rates increased, and species richness decreased over time in both the overall positive and negative groups. These findings also suggest that the system’s biodiversity is no longer increasing, and that the microcosm system helped us to measure species richness and extinction rates by excluding the impacts of dispersion and other variables. As a consequence, we have successfully built an experimental system of microbial microcosms that satisfies the prerequisites for testing the resource hypothesis, as well as the predetermined goals.

However, greater resource abundance in both treatment groups did not result in higher population densities (i.e., relative microbial richness) for each taxon. There was also no significant negative correlation between extinction rate and total resource, nor between extinction rate and volume. Resources did not change the structure or complexity of the microbial interaction network in the microcosm. These results imply that although the present study design has excluded many influences and ensured the premise that larger habitats have more resources, resources did not shape microbial SAR through extinction in the present experimental system. In contrast to cryoconite holes on less productive sites, Sommers et al. (2020) discovered that cryoconite holes on more productive locations accumulated more species with increasing hole area [40]. Productivity, however, was unable to account for all the additional variance in the SAR. A cryoconite hole is a natural micro-ecosystem, and in addition to production (which also represents resources), there are numerous other factors that influence SAR. By excluding these additional factors from the experiment in this study, it was shown that resources cannot independently shape microbial SAR.

Based on insect population ecology research, some researchers believe that more resources will entice more insects to move into the area [13,41]. According to this theory, more resources in a larger habitat will attract more species to move in, giving the resources the ability to shape the SAR. According to Connor and McCoy (2013), the resource hypothesis appears to be more effective in bird and insect populations [11]. Because of their great dispersal capacity, birds and insects may be more susceptible to the resource hypothesis than other species. Therefore, the SAR generation process of microorganisms with a high dispersing capacity may be also well explained by the resource hypothesis. To explore the influence of resources on dispersal and to confirm the SAR resource hypothesis, microorganisms are key.

In essence, biodiversity is determined by only three processes, speciation, dispersal, and extinction [42]. The SAR, which defines the link between regional biodiversity and area, can also be shaped by these three essential processes. By starting with extinction, our study was able to thoroughly test the resource hypothesis and rule out the possibility that it altered SAR by altering extinction rates.

The future of SAR, and indeed biodiversity research in general, should return to the three key processes, for which microbial microcosm experiments are uniquely suited. Firstly, microcosm experiments enable continuous monitoring of temporal changes in microbial diversity. Studies have found that Escherichia coli can reproduce one generation every 20–30 min. In the 30-day microcosm experiment of this study, the microorganisms may have reproduced for more than 2000 generations. From the perspective of humans, the 2000 generations have spanned 50,000 years of changes (humans are calculated as 1 generation in 25 years). Only microbial microcosmic systems are capable of conducting research with such a large generation span. Secondly, the microcosmic system is able to independently verify the individual factors that influence microbial diversity. The dilemma with current research on SAR mechanisms is that the factors that influence species richness in natural habitats are often indistinguishable from each other [43]. For example, an increase in habitat size implies an increase in habitat diversity and resource availability. To overcome this dilemma, we must exclude the role of all other influences when verifying the role of one factor, and we must use the microcosm system. Thirdly, microbial microcosm experiments allow for quantitative monitoring of individual ecological variables. This is also difficult to achieve in natural ecosystems.

## 5. Conclusions

Based on the results, we believe that resources cannot independently shape the SAR through extinction; however, we cannot rule out the possibility that resources shape the SAR through factors such as dispersal. We call for future research to further validate this relationship from a dispersal perspective.

## Figures and Tables

**Figure 1 microorganisms-10-01993-f001:**
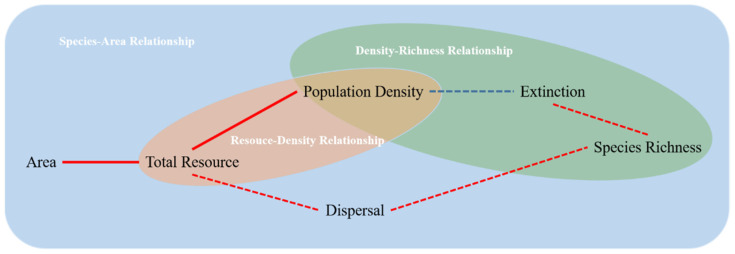
The research framework for this study. The red line segments in the figure indicate increases, blue lines indicate decreases, solid lines indicate proven relationships and dashed lines indicate relationships that are controversial. The blue boxed area represents the mechanism of resource-related SAR formation, the red oval area represents the mechanism for the resource–density relationship, and the green oval area represents the mechanism for the density–richness relationship. The resource hypothesis would be confirmed as one of the SAR formation mechanisms if the population density of each species increases when the total resource increases with area, which in turn leads to a lower extinction rate.

**Figure 2 microorganisms-10-01993-f002:**
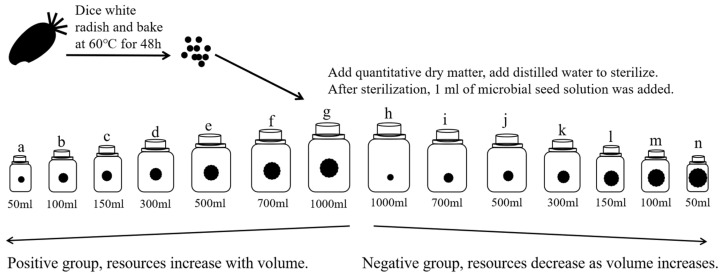
Schematic diagram of the experimental design. a–n is the id of each experimental group.

**Figure 3 microorganisms-10-01993-f003:**
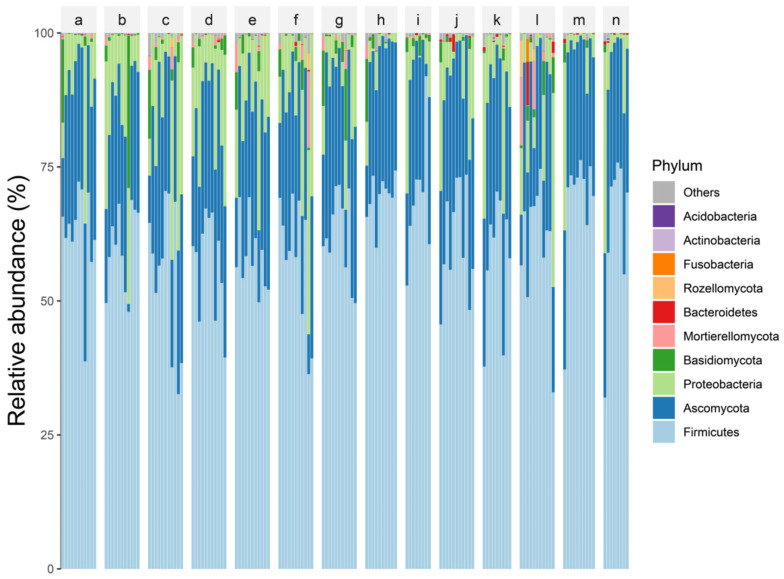
Histogram of dominant phylum stacking. A–n are the treatment group numbers and the top 10 phyla in terms of relative abundance of fungi and bacteria are shown. a–n is the id of each experimental group of Figure 2.

**Figure 4 microorganisms-10-01993-f004:**
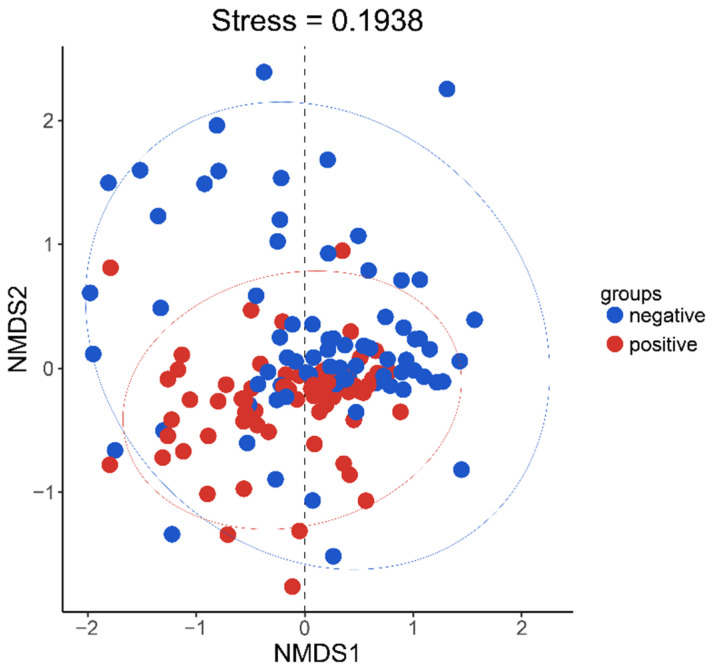
NMDS analysis of the two treatment groups. Positive is the treatment where dry matter mass is positively related to volume; negative is the treatment where dry matter is negatively related to volume.

**Figure 5 microorganisms-10-01993-f005:**
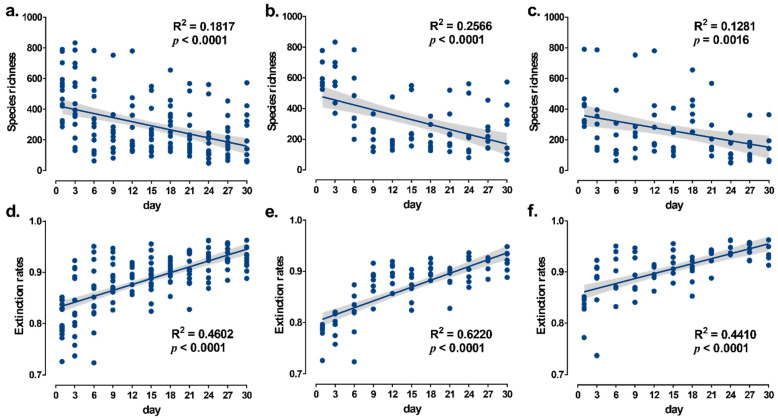
Linear relationships between microbial species richness and extinction rates in the microcosm system and time. (**a**–**c**) are linear relationships between species richness and microcosm establishment time for the overall positive and negative groups, respectively; (**d**–**f**) are the linear relationships between extinction rates and microcosm establishment time for the overall positive and negative groups, respectively. The solid line in the figure shows the linear regression model and the grey shading is the 95% confidence interval.

**Figure 6 microorganisms-10-01993-f006:**
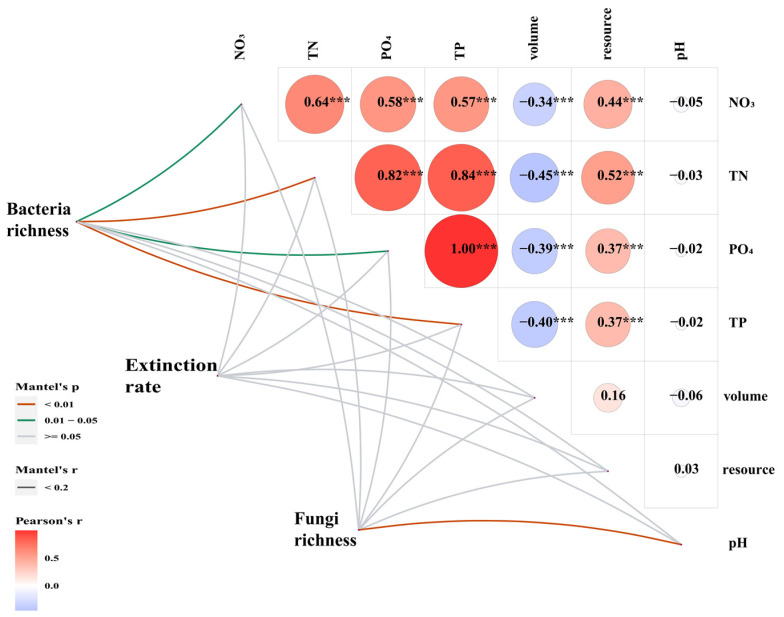
Mantel test correlation plot. The heat map shows the Pearson correlation between the environmental factors; the thickness of the line represents the size of the Mantel test correlation coefficient; the color of the line indicates the *p*-value of the Mantel test. * p < 0.05, ** p < 0.01, *** p < 0.001.

**Figure 7 microorganisms-10-01993-f007:**
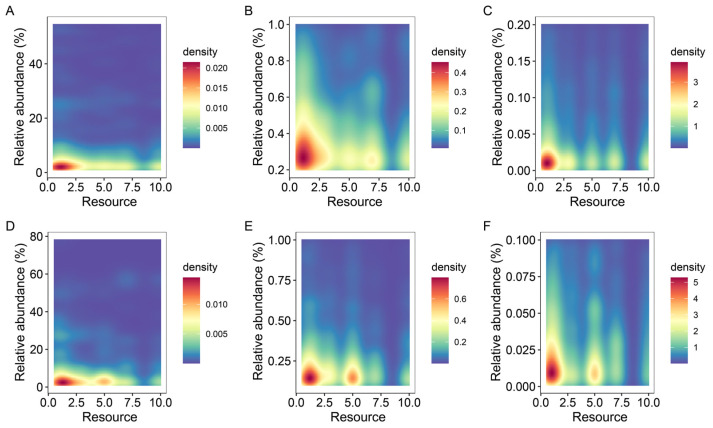
Kernel density diagram of the relationship between resources and relative abundance of microorganisms (fungi and bacteria). In this figure, (**A**–**C**) are positive groups and (**D**–**F**) are negative groups; (**A**,**D**) are microbial taxa with relative abundance greater than 1%, (**B**,**E**) are microbial taxa with relative abundance between 0.1% and 1%, and (**C**,**F**) are microbial taxa with relative abundance less than 0.1%. Density is the kernel density of the number of species with corresponding relative abundance at a locus in the figure.

**Figure 8 microorganisms-10-01993-f008:**
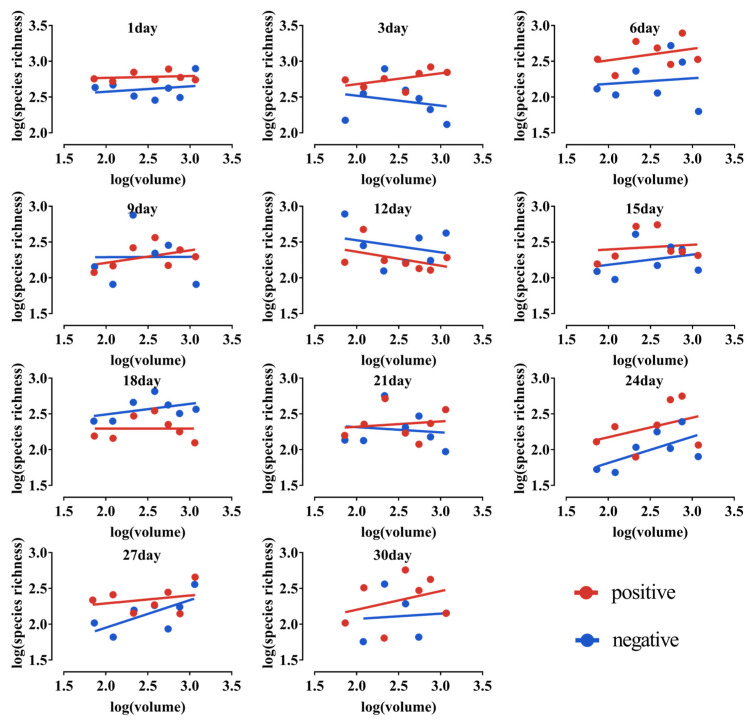
SAR curves for the microcosm system over 30 days. Positive is the treatment where dry matter mass is positively related to volume; negative is the treatment where dry matter is negatively related to volume. The solid line is the regression line.

**Figure 9 microorganisms-10-01993-f009:**
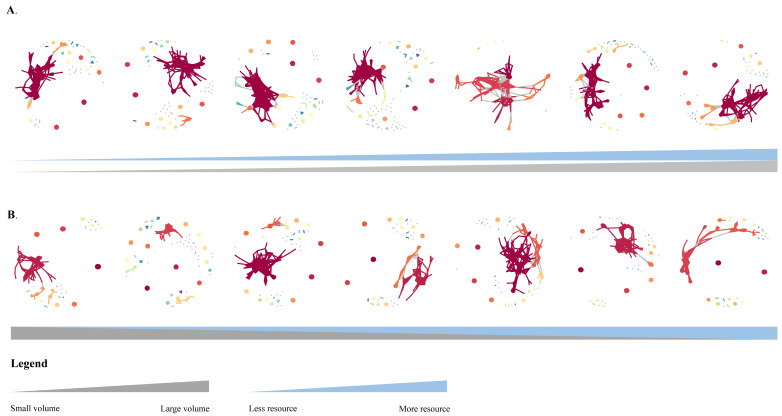
Microbial interactions network. (**A**) is the positive group, (**B**) is the negative group; grey triangular areas represent microcosm volume changes and blue triangular areas represent resource volume changes.

**Figure 10 microorganisms-10-01993-f010:**
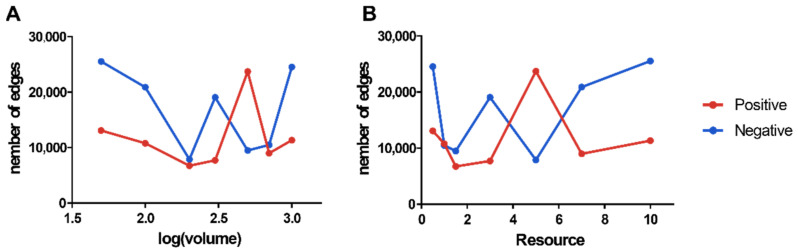
Plot of the number of microbial interactions network edges versus microcosm volume and resources. (**A**) shows the number of interactions network edges versus microcosm volume; (**B**) shows the number of interactions network edges versus resources.

**Table 1 microorganisms-10-01993-t001:** Statistical results of SAR curve fitting.

	Positive			Negative		
Day	R Square	Slope	*p*	R Square	Slope	*p*
1	0.0318	0.0258	0.7022	0.0484	0.0756	0.6355
3	0.2959	0.1521	0.2068	0.0514	−0.1391	0.6251
6	0.1200	0.1618	0.4466	0.0117	0.0780	0.8171
9	0.2010	0.1754	0.3130	<0.0001	0.0043	0.9907
12	0.2008	−0.1952	0.3133	0.0714	−0.1691	0.5624
15	0.0181	0.0660	0.7734	0.0760	0.1428	0.5496
18	<0.0001	0.0002	0.9992	0.1901	0.1485	0.3282
21	0.0218	0.0746	0.7523	0.0148	−0.0734	0.7947
24	0.1406	0.2738	0.4072	0.3824	0.3676	0.1388
27	0.0734	0.1107	0.5568	0.4594	0.3844	0.0943
30	0.1059	0.2606	0.4764	0.0071	0.0742	0.8928

## Data Availability

The data presented in this study are available upon request from the corresponding author.

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
