# Peer review of "Testing the Resource Hypothesis of Species–Area Relationships: Extinction Cannot Work Alone"

_microorganisms, 2022, doi:10.3390/microorganisms10101993_

Round 1

Reviewer 1 Report

Comment 1: Water is an important resource for microbial growth. Is water considered a resource in microbial experimental design? This article seems to only consider dry matter as a resource.

Comment 2: Please note that strain names mentioned in the article should be in italics. For example line 219.

Comment 3: In Figure 3, which treatments do groups a-n represent?

Comment 4: Is sequencing third generation or second generation? In the analysis, only genus- and phylum-level microbial structures appeared to be of interest.

Comment 5: In this study, was microbial extinction defined as the complete disappearance of a microbial genus?

Comment 6: The closed experimental environment prevents other microorganisms from entering the "experimental universe". Will this affect the expected study results? 313-322 Road

评论1:水是微生物生长的重要资源。在微生物实验设计中,水被认为是一种资源吗?这篇文章似乎只考虑将干物质作为一种资源。

评论2请注意,文章中提到的菌株名称应为斜体。比如第219行。

评论3:在图3中,a-n组代表哪些治疗方法?

评论4:测序是第三代还是第二代?在分析中,似乎只有属和门级微生物结构被关注。

评论5:在这项研究中,微生物灭绝是否被定义为微生物属的完全消失?

评论6:封闭的实验环境阻止了其他微生物进入“实验宇宙”。这会影响预期的研究结果吗?313-322路

Author Response

Dear reviewer,

Thank you for your careful review and constructive comments on our article. We have revised the manuscript based on your suggestions. Here are our answers to the questions you raised.

Comment 1: Water is an important resource for microbial growth. Is water considered a resource in microbial experimental design? This article seems to only consider the dry matter as a resource.

Response: Although water is a resource for microbes as well, we do not regard it as a limited resource in our research because it is present in large quantities in all microcosmic systems. More importantly, microcosm volume in this study was positively correlated with water content, but the results of the study showed no correlation between volume and extinction rate. Therefore water content also did not affect the formation of SAR in this study.

Comment 2: Please note that strain names mentioned in the article should be in italics. For example line 219.

Response: We have made formatting changes as you suggested.

Comment 3: In Figure 3, which treatments do groups a-n represent?

Response: a-n is the id of each experimental group in Figure 2. We have added the corresponding elucidations in the legend of Figures 2 and 3, respectively.

Comment 4: Is sequencing third generation or second generation? In the analysis, only genus- and phylum-level microbial structures appeared to be of interest.

Response: We applied a second-generation sequencing technique in this investigation. Even though we used higher taxonomic resolution ASVs rather than OTUs for diversity analysis, many of the ASVs still had poor levels of species annotation, which is a problem that much recent research faces. As a result, in the diversity overview part of our study, we primarily presented data at the phylum and genus levels. As for the SAR analysis, we calculated the richness at the ASV level.

Comment 5: In this study, was microbial extinction defined as the complete disappearance of a microbial genus?

Response: We define the complete disappearance of an ASV is recorded as extinction. We added this definition to the data analysis section.

Comment 6: The closed experimental environment prevents other microorganisms from entering the "experimental universe". Will this affect the expected study results? 313-322 Road.

Response: Indeed, the closed microcosmic system prevents the entry of other microorganisms. This study's approach was intended to eliminate the impact of dispersal on SAR and so independently confirm the contribution of resources. The results lead us to assume that resources cannot, on their own, influence SAR through extinction, but we cannot completely rule out the potential that resources influence SAR through elements like dispersal. We urge more studies to confirm this connection from a dispersal standpoint.

We appreciate your warm work earnestly and hope that the correction will meet with approval. Once again, thank you very much for your comments and suggestions.

All the best,

Deng Wei

Reviewer 2 Report

The authors present a manuscript in which they explore the concept SAR in the experiment with a microbial community.

This approach makes it possible to study all the regularities in controlled experiments and obtain results for comprehensive statistical processing. Nevertheless, it should be taken into account that the development of a microbial community as a system differs essentially from that of plant and animal communities, which were the basis for the above SAR concept.

The system, which the authors name the microcosm, can be better defined as a fermentation process in which a microbial seed is added to a substrate in order to start fermentation. The system can vary greatly depending on various factors such as:

a) the substrate degradability that can favor selectively only some microbial groups

b) inhibition of fermentation at a high concentration of the substrate

c) low rate of the process when the available substrate or mineral components are limited

Usually, the microcosm for microbiologists is a system containing the already existing substrate material and the native microbial community, which is studied without introducing an additional microbial association into it. Such a microcosm is a relatively balanced system, the potential of which can be studied by varying different parameters. In the presented work, the system is significantly unbalanced and comes to balance gradually. Thus, what is being studied is not the response of a balanced system to various parameters, but the process of gradually establishing a balance in a completely new system.

If we are taking all of the above into account, then it is important to clarify whether such an initially unbalanced system, used by the authors in the work, can be applied to the study of the SAR concept? Is it possible to level out the factors associated with the unbalanced microbial system and its possible connection to excess or limiting parameters, talking about SAR?

Authors should carefully analyze all the factors associated with the microbial fermentation process and establish the proper relevance of such a system and the main goal of the study. They may need to consult with a microbiologist or biotechnologist!

In addition, Fig. 1 in Materials and Methods should be changed to Fig. 2.

Author Response

Dear reviewer,

We appreciate your thoughtful analysis and helpful criticism of our paper. Here are our answers to the questions you raised.

First of all, you suggested that plant and animal communities are substantially different from microbial communities as a whole. Additionally, you suggested us drawing attention to the fact that the experimental system is a system of fermentation processes, which is an imbalanced system, and you expressed uncertainty as to whether it can be utilized for SAR investigations.

Indeed, our microcosmic system is a fermentation process. But from the ecological point of view, the fermentation process is the process of organisms using the resources in the habitat to build communities, achieve community functions, and maintain biodiversity. This process of community building is similar to that of plant and animal communities, except that it happens and evolves more quickly, In reality, community succession in all taxa occurs gradually; the idea that there is a top pattern or endpoint is still debatable. In a different research, we discovered that two processes—the decrease of decomposable substrates and the suppression of fermentation by high substrate concentrations in fermentation systems—were essentially what pushed the system in the direction of stability. We observed significant SAR in that study, proving it is possible to investigate the SAR process in this research system.

You also mention that some microbial taxa are favored during fermentation according to the substrate's tendency to degrade.

The development of ecological niche differences among species is one of the requirements for SAR. The various ways that microorganisms in this research system use the same resource result in ecological niche differences among microorganisms in this system.

All the best,

Deng Wei

Reviewer 3 Report

The authors Wei Deng, Lilei Liu, Guo-Bin Yu, Na Li, Yang Xiao-yan and Wen Xiao have submitted a manuscript entitled “Testing the resource concentration hypothesis of species-area relationships: extinction cannot work alone” to the scientific journal: Microorganisms (Manuscript ID: microorganisms-1920975). After a minor revision this draft which describes a solid piece of scientific work will become a valuable article for the general readership of the journal Microorganisms. To my mind it is possible to discuss the reported results even in a bit broader context. Therefore, I have listed below three additional references which te authors may discuss in their article:

John M. Halley, Vasiliki Sgardeli, Nikolaos Monokrousos (2013) Species–area relationships and extinction forecasts. The Year in Ecology and Conservation Biology Volume 1286, Issue 1. Pages: 50-61. https://doi.org/10.1111/nyas.12073

Guochun Shen, Mingjian Yu, Xin-Sheng Hu, Xiangcheng Mi, Haibao Ren, I-Fang Sun and Keping Ma (2009) Species-Area Relationships Explained by the Joint Effects of Dispersal Limitation and Habitat Heterogeneity. Ecology Vol. 90, No. 11, pp. 3033-3041. https://www.jstor.org/stable/25592844

Sommers P, Porazinska DL, Darcy JL, Gendron EMS, Vimercati L, Solon AJ, Schmidt SK (2020) Microbial Species-Area Relationships in Antarctic Cryoconite Holes Depend on Productivity. Microorganisms. 2020 Nov 7;8(11):1747. doi:10.3390/microorganisms8111747. PMID: 33171740; PMCID: PMC7694949.

Author Response

Dear reviewer,

We appreciate your thoughtful analysis and helpful suggestions of our paper. Based on your recommendations, we changed the manuscript, greatly improving the discussion part. We can illustrate the viability of our experimental design with the aid of Shen et al. (2009) and Halley et al. (2013) findings, which also strongly corroborate our discussion of extinction and dispersal. The discussion of the resource hypothesis is well supported by the findings of Sommers et al. (2020).

Our research team is currently testing each hypothesis of SAR formation one by one using the microcosmic system. We would like to invite you to stay tuned and discuss these findings with us at any time!

All the best,

Deng Wei

Round 2

Reviewer 2 Report

Dear authors,

your point of view is understandable. There are still some discussible points, but your information about other experiments makes sure that such study will be continued!

Reviewer 3 Report

The authors have followed all suggestions.